# Roles of JNK/Nrf2 Pathway on Hemin-Induced Heme Oxygenase-1 Activation in MCF-7 Human Breast Cancer Cells

**DOI:** 10.3390/medicina56060268

**Published:** 2020-05-29

**Authors:** Hye-Yeon Jang, On-Yu Hong, Eun-Yong Chung, Kwang-Hyun Park, Jong-Suk Kim

**Affiliations:** 1Department of Biochemistry and Institute of Cardiovascular Research, Chonbuk National University Medical School, Jeonju 54896, Korea; janghyeyeon@naver.com (H.-Y.J.); khong9053@naver.com (O.-Y.H.); 2Department of Anesthesiology and Pain Medicine, Bucheon St. Mary’s Hospital, Catholic University of Korea, Bucheon 14647, Korea; anes36@catholic.ac.kr; 3Department of Emergency Medical Rescue, Nambu University, Gwangju 62271, Korea; 4Department of Emergency Medicine, Graduate School of Chonnam National University, Gwangju 61469, Korea

**Keywords:** heme oxygenase-1, JNK, Nrf2, breast cancer, MCF-7, hemin, brazilin

## Abstract

Heme oxygenase-1 (HO-1) is highly induced in various human disease states, including cancer, indicating that HO-1 is an emerging target of cancer therapy. In this study, we investigated that the mechanisms of hemin-induced HO-1 expression and its signaling pathways in human breast cancer cell. We used MCF-7 cells, a human breast cancer cell line. Hemin increased HO-1 expression in MCF-7 cells in a dose- and time-dependent manner. Hemin enhanced HO-1 expression through the activation of c-Jun N-terminal kinases (JNK) signaling pathway. Hemin also induced activation of Nrf2, a major transcription factor of HO-1 expression. These responses in MCF-7 cells were completely blocked by pretreatment with brazilin, a HO-1 regulator. These results indicated that brazilin inhibits hemin-induced HO-1 expressions through inactivation of JNK/Nrf2 in MCF-7 cells. Thus, our findings suggest that HO-1 is an important anticancer-target of brazilin in human breast cancer.

## 1. Introduction

Heme oxygenase-1 (HO-1) was identified by Maines [1], as a liver microsomal protein with degradation activity of heme to bilirubin. HO-1 catalyzes the degradation of heme to form the open-chain tetrapyrrole biliverdin-Ixα, carbon monoxide (CO), and free iron, which play crucial roles in the adaptation to defense against oxidative stress and cellular stress [2,3]. HO-1 is induced in cells by cellular stress including oxidants [4], hypoxia condition [5], cytokine [6,7], and ultraviolet light irradiation [5,8]. These findings suggest that HO-1 has cytoprotective actions. Therefore, studies focused on HO-1 to find potential managements for various diseases, including cancer [9,10], inflammatory diseases [11], respiratory disease [12], and circulation system [13,14]. It is known that expression of high level of HO-1 occurs in various tumors including in renal cancer [15], acute T cell leukemia [16], prostate cancer [17], lung [18], and squamous cell carcinoma in head and neck [19]. Many studies suggested the HO-1 were implicated in tumorigenesis and cancer prognosis like chemotherapeutic sensitivity [20], cancer invasion [21,22], aggressiveness [23,24] and survival rate prediction [25]. Taken together, these findings indicate that the investigating of the role of HO-1 and its targets with molecular mechanism for cancer therapy seem to be important [26].

Breast cancer (BRCA) is most common types of cancer in woman worldwide, and one of the most important causes of cancer-related mortality for women [27,28]. MCF-7 cells were obtained at the Michigan Cancer Foundation (MCF) in 1973 and most commonly used xenograft model of breast cancer [28]. In gene expression profiling analysis, the MCF-7 were possess characteristics that estrogen receptor 1 (ESR1)-positive, progesterone receptor (PGR)-negative, Erb-B2 Receptor Tyrosine Kinase 2 (ERBB2/HER2)-negative and epidermal growth factor receptor (EGFR)-negative [29]. Whereas heme oxygenase (HOs), CCAAT/enhance binding protein (C/EBP) homolog protein (CHOP), glucocorticoid-inducible kinase 1 (SGK-1), prostate apoptosis response (Par-4), Caveolin-1 (Cav-1) mRNA expressions affects to cellular survivals [27]. Therefore, there are requires studies how heme oxygenase affects cancer progression/regulation in MCF-7 cells, the molecular mechanism of playing role in the search of anti-cancer technology.

*Caesalpinia sappan* (*C*. *sappan*) has been used as an Asian traditional medicine for acute/chronic human diseases [30]. The major component of *C*. *sappan*, 7,11b-dihydrobenz[b]indeno [1,2-d] pyran-3,6a,9,10(6H)-tetrol (brazilin) [30,31], exhibits various biological effects, including anti-inflammatory [32], and anti-hepatotoxicity activity [33], and inhibition of protein kinase C [34]. Recent studies have reported that brazilin has anticancer activities such as anti-mitotic and apoptotic activity [35,36]. In this study, we examined whether brazilin induces expression of HO-1 using a human breast cancer line.

## 2. Materials and Methods

### 2.1. Cells and Materials

A breast cancer cell line, MCF-7, was purchased from American Type Culture Collection (ATCC; VA. USA). The cells were maintained in high glucose containing DMEM (Dulbecco’s modified Eagle’s medium)–10% fetal bovine serum (FBS, *v*/*v*)–antibiotics/antimycotics (Gibco, Gaithersburg, MD, USA) at CO_2_ incubator. Brazilin was obtained from MP Biomedicals LLC (Irvine, CA, USA). Hemin, 3-(4,5-dimethyl-thiazol-2-yl)-2,5-diphenyltetrazolium bromide (MTT), DMEM, and anti-β-actin antibody were purchased from Sigma-Aldrich (St. Louis, MO, USA). The antibodies related to phospho-JNK (p-JNK, Cat #4668), phospho-p38 (p-p38, Cat #9216) and phospho-ERK (p-ERK, Cat #4695) were purchased from Cell Signaling Technology (Beverly, MA, USA). HO-1 antibody was from Enzo Life Sciences (Cat #ADI-OSA-111-D, NY, USA). Nrf2 (Cat # sc-13032), PCNA (Cat # sc-9857), and secondary antibodies were from Santa Cruz Biotechnology (Paso Robles, CA, USA). Specific inhibitors against MAPK, SP600125 (JNK inhibitor), SB203580 (p38 inhibitor) and PD98059 (ERK inhibitor) were purchased from MilliporeSigma (Burlington, MA, USA). [α-^32^P]dCTP was purchased from Amersham plc (Buckinghamshire, UK). Culture medium, supplements and cell culture tested PBS were obtained from Gibco (ME, USA).

### 2.2. Cell Viability Assay

The cell viability (and/or growth rate) of MCF-7 was assayed using an MTT assay. Briefly, cells were inoculated in a microtiter plate 96 well (3 × 10^3^ cells/well, confluent less than 60%), and then incubated at CO_2_ incubator for 12 h. Cells were either untreated or treated with 2.5, 5, 10, 20, 50 μM brazilin and 25, 50, 100 μM hemin at 37 °C and further incubated for 24 h. The cells were washed with PBS prior to the addition of 0.5 mg/mL MTT (100 μL per well), and then incubated at 37 °C for 30 min. Insoluble formazan were solubilized with DMSO (100 μL per well) and measured at 570 nm using a microplate reader (Model 3550, Bio-Rad, Hercules, CA, USA).

### 2.3. Western Blot Analysis

MCF-7 cells (7.0 × 10^5^ cells/sample) were treated with 2.5, 5, 10, 20 μM brazilin for 60 min, and then incubated with 50 μM hemin at 37 °C for 24 h. Cells were lysed with commercial ice-cold lysis reagents (M-PER Mammalian Protein Extraction Reagent, Pierce Biotechnology, Rockford, IL, USA). Protein extracts (10 μg per well) were separated by SDS-PAGE and then transferred to polyvinylidene fluoride (PVDF) membranes (GE Healthcare Life Sciences, Pittsburgh, PA, USA). Membrane was blocked for 1 to 2 h with 5% skim milk or 3% immunoglobulin-free bovine serum albumin (BSA), and then incubated 15 h at 4 °C with 1.0 μg/mL of primary antibody (1:1000 to 2000). Horseradish peroxidase-conjugated anti-IgG monoclonal antibody (1:1000 to 2000) was used as the secondary antibody. Specific band were analyzed with LAS-1000 (Fuji Film, Tokyo, Japan). The images were quantified using a computer image analysis software, Image Reader Pro (Fuji Film, Tokyo, Japan) and ImageJ (National Institutes of Health, Bethesda, MD, USA).

### 2.4. Quantitative Real-Time Polymerase Chain Reaction

Total RNA was isolated using the commercial kits (FastPure^TM^ RNA Kit, Takara Bio Inc., Kusatsu, Shiga Japan). The RNA concentration and purities were determined by absorbance at 260/280 nm. cDNA was synthesized from 1 μg total RNA using a PrimeScript^TM^ RT reagent Kit (Takara, Japan). HO-1 and glyceraldehyde 3-phosphate dehydrogenase (GAPDH) mRNA expression were determined by real-time qPCR using the sequence detection system (ABI PRISM 7900, Applied Biosystems, Waltham, MA, USA), and SYBR Green. The primers were used to amplify cDNA of HO-1 (NM002133) (sense: 5′-GCC AGC AAC AAA GTG CAA G-3′; antisense: 5′-GGC ATA AAG CCC TAC AGC AA-3′) and GAPDH (NM002046) (sense: 5′-ATG GAA ATC CCA TCA CCA TCT T; antisense: CGC CCC ACT TGA TTT TGG-3′) (Genotech, Daejeon, Korea). The variation in mRNA concentration of all genes were normalized to the GAPDH housekeeping gene. Relative quantitation was performed using the comparative Ct method according to the manufacturer’s instructions. Data were presented the mean ± S.E.M of three independent experiments.

### 2.5. Isolation Extracts of Nuclear Fraction

MCF-7 cells (2 × 10^6^ cells/sample) were treated with brazilin in the presence or absence of hemin (50 μM) for 4 h. Cells were washed twice and carefully detached by scrapping into 1.5 mL of ice-cold PBS (pH 7.5) and then pelleted (4000× *g* at 4 °C for 5 min). Cytoplasmic and nuclear fraction were extracted from cells using commercial kit (NE-PER)(Pierce Biotechnology, IL, USA).

### 2.6. Electrophoretic Mobility Shift Assay (EMSA)

Nuclear extract of cells was prepared as described above. An oligonucleotide containing the Nrf2 (5′-TGG GGA ACC TGT GCT GAG TCA CTG GAG-3′) were purchased from Genotech (Daejeon, Korea). Electrophoretic mobility shift assay was performed as described in Hellman and Fried [37]. Specific binding was controlled by competition with a 50-fold excess of cold Nrf2 oligonucleotide.

### 2.7. Statistical Analysis

All tests were performed at least in three independent experiments. To compare means within groups we performed student *t*-test. Values of *p* < 0.05 were considered statistically significant.

## 3. Results

### 3.1. Endogenous Cytotoxicity of Brazilin and Hemin on MCF-7 Cells

To examine the endogenous cytotoxicity of brazilin (Figure 1A) and hemin (Figure 1B) on MCF-7 cells, we treated cells with brazilin (0–50 μM) and hemin (0–100 μM) for 24 h. The influence of brazilin and hemin on MCF-7 cellular toxicity was then analyzed using MTT assay. Treatment of MCF-7 cells with indicated dosage of brazilin and hemin shown no significant change in cell viability (Figure 2A,B). Therefore, we performed further experiments in a non-toxic dose of brazilin and hemin.

### 3.2. Effect of Hemin on HO-1 Expression in MCF-7 Cells

To investigate the effect of hemin on HO-1 expression in MCF-7 cells, we performed Western blotting. Western blot analysis revealed that hemin increased HO-1 expression in a dose- and time-dependent manner (Figure 3A,B).

### 3.3. Effect of Brazilin on Hemin-Induced HO-1 Expression in MCF-7 Cells

To investigate the effect of brazilin on hemin-induced HO-1 expression in MCF-7 cells, we performed Western blotting and real-time PCR. Western blot analysis and real-time PCR revealed that brazilin-suppressed hemin-induced HO-1 expression in a dose-dependent manner (Figure 4A,B).

### 3.4. Effects of Hemin on MAPK Activation in MCF-7 Cells

The effect of hemin on JNK/ERK/p38 activities was investigated using Western blotting. MCF-7 cells pretreated with inhibitors of ERK (PD98059), JNK (SP600125), and p38 (SB203580) for 1 h were further incubated with hemin (50 μM) for 24 h. As shown in Figure 5A,B, inhibition of JNK only blocked hemin-induced HO-1 protein expression in MCF-7 cells. These results suggest that hemin-induced HO-1 expression were mediated by JNK signal pathway.

### 3.5. Effects of Brazilin on Hemin-Induced JNK Activation in MCF-7 Cells

To investigate the effect of brazilin on hemin-induced JNK activation, we performed Western blotting. As shown in Figure 6A–D, hemin significantly induced phosphorylation of JNK only. Interestingly, brazilin also only significantly blocked phosphorylation of JNK by hemin in MCF-7 cells (Figure 6A,D).

### 3.6. Effect of Brazilin on Hemin-Induced Nrf2 Activation in MCF-7 Cells

To further understand the inhibitory mechanism of brazilin on HO-1 transcriptional regulation, we determined whether brazilin inhibits Nrf2 activation in MCF-7 cell after stimulation with hemin using Western blotting and EMSA. Brazilin caused inhibitory actions on nuclear translocation of Nrf2 (Figure 7A–C). Furthermore, DNA binding activity of hemin-induced Nrf2 was markedly inhibited by treatment of brazilin in a dose-dependent manner *(*Figure 8A,B). These results indicate that Nrf2 nuclear translocation is an important mechanism on hemin-induced HO-1 expression and that brazilin regulates Nrf2 translocation.

## 4. Discussion

This study provides evidence that brazilin, a major component of *Caesalpinia sappan*, blocks hemin-induced HO-1 expression in MCF-7 cells and JNK/Nrf2-mediated HO-1 expression. Thus, HO-1 may be an important target in the screening of novel pharmaceuticals to treatment of breast cancer. In human cells, two isoforms of HO are present. HO-1 is a stress-inducible enzyme, while HO-2 is a constitutive enzyme as genetically and enzymatically [38]. Hemin induces the expression of HO-1 in a variety of cell types including endothelium [39], aortic tissues [40], embryonic fibroblast and adipocytes [41]. 

In this study, hemin increased HO-1 expression in MCF-7 cells in a dose- and time-dependent manner, and brazilin substantially inhibited hemin-induced HO-1 expression in human breast cancer cells. It has been known that MAPK and ROS control HO-1 expression through Nrf2 activation in various cells such as pulmonary aortic endothelial cells (PAEC) [42], hepatic and hematopoietic system [43], and hepatic carcinoma cells [44]. Indeed, increases of HO-1 expression were supported anti-apoptotic effects in monocyte [45], tumor necrosis [46], then many trials were suggested with previously reported inhibitors [45]. Therefore, this study reveals that an increase in the level of hemin-induced HO-1 markedly reduces the signaling mechanism of MCH-7 cells by brazilin.

In this study, hemin increased HO-1 expression (protein and mRNA level) in MCF-7 cells in a dose- and time-dependent manner. Among MAPK signaling molecules, hemin induced only JNK activation in MCF-7 cells. Our results also showed that hemin induces HO-1 expression through Nrf2 activation in MCF-7 cells. Interestingly, these responses were completely blocked by pretreatment with brazilin, indicating that brazilin inhibits hemin-induced HO-1 expressions through inactivation of JNK/Nrf2 in MCF-7 cells. In cancer cells, elevation of HO-1 expression increased in various tumors [15,16,17,18,19] and hemin, an iron-containing porphyrin with chlorine, is available HO-1 inducer in various cell types including normal and cancer cells [47,48]. In this study, we observed hemin-induced increases of HO-1 expression in MCF-7 cells and brazilin is potent regulator through HO-1 expression. On the other hand, we investigated whether brazilin-induced HO-1 expression, but it was not observed (data not shown). Moreover, novel HO-1 inhibition tools were reported such as arylethanolimidazole derivatives in cancer [49], caffeic acid phenethyl ester in diabetes [50] and fumarate hydratase as target genes in cancer [51]. 

Taken together, these data and previous publications indicates that hemin induced HO-1expression and JNK/Nrf2 signaling pathway plays a regulatory role in MCF-7 cells indicates that cytoprotective role in MCF cells. Therefore, blockage effects of brazilin against hemin-induced HO-1 expression in MCF-7 cells suggest useful therapeutic/prevention implication. However, the brazilin growth-inhibitory effect on human breast cancer cells still need to be studied.

This study was concluded that HO-1 is an important anti-cancer target of brazilin in human breast cancer via inhibit hemin-induced HO-1 expression through inactivation of JNK/Nrf2 in MCF-7 cells [52]. Therefore, brazilin or other HO-1 regulators may be of use in the treatment breast cancer and required further studies.

## Figures and Tables

**Figure 1 medicina-56-00268-f001:**
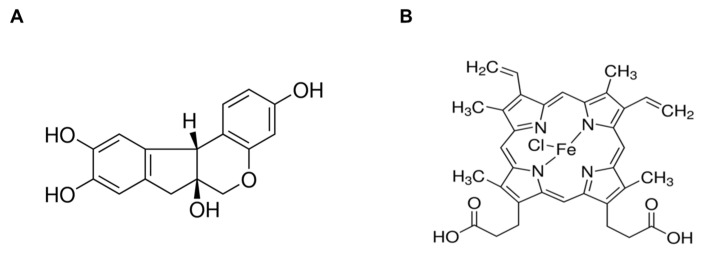
Chemical structure of (**A**) brazilin (CAS number: 474-07-7, C_16_H_14_O_5_, MW: 286.28) and (**B**) hemin (CAS number: 16009-13-5, C_34_H_32_ClFeN_4_O_4_, MW: 651.94).

**Figure 2 medicina-56-00268-f002:**
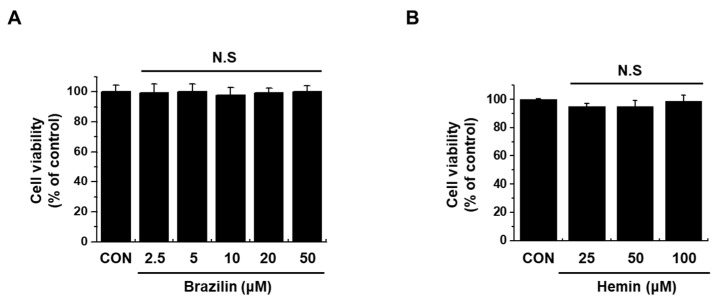
Effect of brazilin and hemin on the viability of MCF-7 cells. (**A**) Brazilin and (**B**) Hemin was treated with indicated concentration for 24 h. An established MTT assay was used to detect viability of cells. Data are presented as Means ± S.E.M of three independent experiments. N.S stands for not statistically significant.

**Figure 3 medicina-56-00268-f003:**
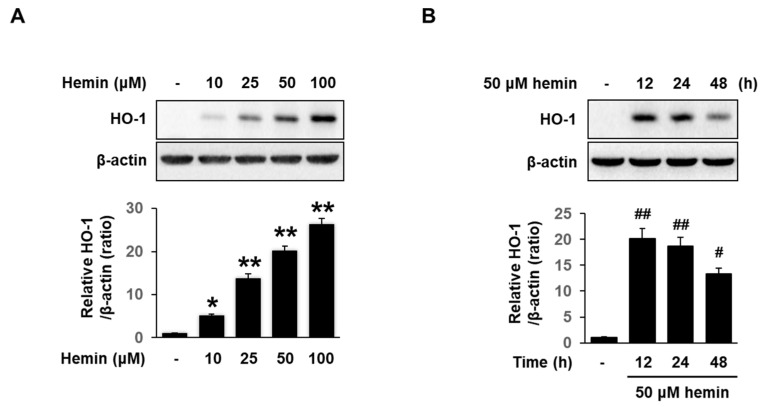
Hemin increase expression of HO-1 in MCF-7 cells. HO-1 protein expression were analyzed by Western blotting. (**A**) MCF-7 cells were treated with 10, 25, 50, and 100 μM hemin for 24 h. (**B**) Cells were treated with 50 μM hemin for 12, 24, and 48 h. Lower panels: Quantification of intensity in three different experiments using a ImageJ software. Data are presented as Means ± S.E.M of three independent experiments. * *p* < 0.05, ** *p* < 0.001, ^#^
*p* < 0.01 and ^##^
*p* < 0.001 vs. control.

**Figure 4 medicina-56-00268-f004:**
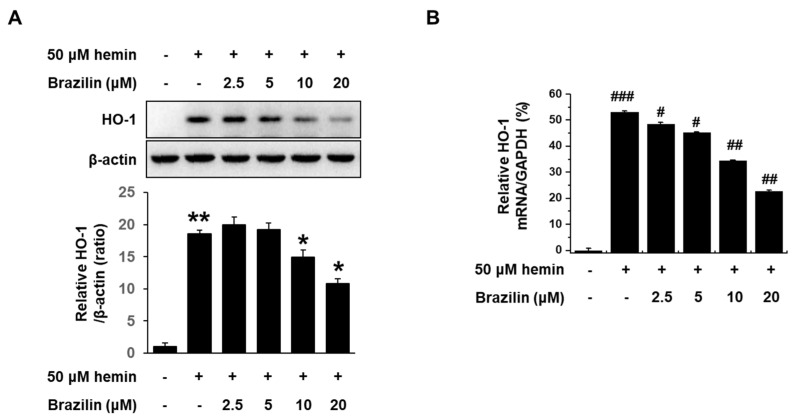
Brazilin regulates increase of hemin-induced HO-1 expression in MCF-7 cells. Regulatory effects of brazilin on (**A**) hemin-induced HO-1 protein expression and (**B**) levels of HO-1 mRNA. Lower panels: Quantification of intensity in three different experiments using a ImageJ software. Cells were treated with brazilin for 1 h, and then incubated with 50 μM hemin for 24 h. HO-1 mRNA levels were analyzed by real-time PCR, and GAPDH was used as an internal control. Data are presented as Means ± S.E.M of three independent experiments. * *p* < 0.05 vs. hemin and ** *p* < 0.001 vs. control. ^#^
*p* < 0.05 and ^##^
*p* < 0.01 vs. hemin. ^###^
*p* < 0.001 vs. control.

**Figure 5 medicina-56-00268-f005:**
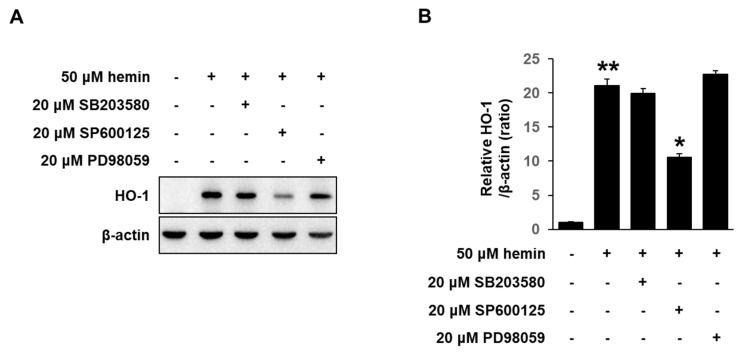
Hemin-induced HO-1 activation through JNK/Nrf2 signaling pathway in MCF-7 cells. (**A**) MCF-7 cells were treated with inhibitors of p38 (20 μM SB203580), JNK (20 μM SP600125) and ERK (20 μM PD98059) for 1 h, and then incubated with 50 μM of hemin for 24 h. The HO-1 protein expression was analyzed using Western blotting. (**B**) Quantification of intensity using a ImageJ software. Data are presented as Means ± S.E.M of three independent experiments. * *p* < 0.01 vs. hemin and ** *p* < 0.001 vs. control.

**Figure 6 medicina-56-00268-f006:**
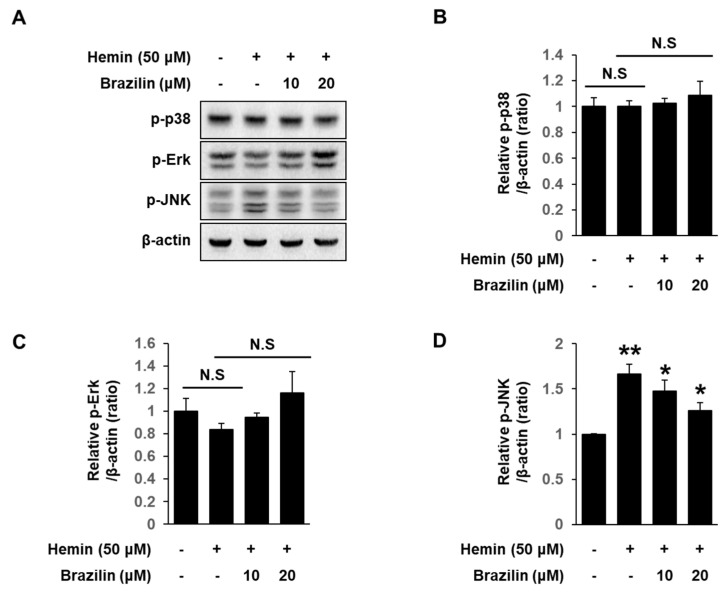
Brazilin regulates hemin-induced HO-1 activation through phosphorylation of Erk and dephosphorylation of JNK. (**A**) Western blot analysis of phosphorylated proteins of p38, Erk and JNK on hemin-induced HO-1 activation. Cells were pretreated with brazilin and then incubated with 50 μM hemin for 1 h. Levels of phosphorylated p38 (**B**), ERK (**C**) and JNK (**D**) were quantified by ImageJ software. Data are presented as Means ± S.E.M of three independent experiments. * *p* < 0.05 vs. hemin and ** *p* < 0.001 vs. control. N.S stands for not statistically significant.

**Figure 7 medicina-56-00268-f007:**
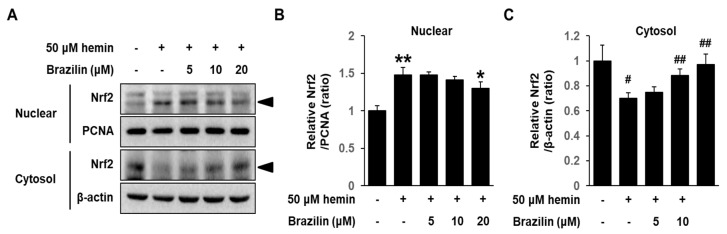
Brazilin regulates hemin-induced Nrf2 translocation into nuclear region. (**A**) Translocation of Nrf2 to the nucleus. Cells were treated with brazilin in the presence of hemin. Following 4 h incubation, nuclear and cytoplasm extracts were prepared. Protein levels were determined by Western blotting. PCNA for nuclear protein and β-actin for cytosol protein were used as loading controls. (**B**,**C**) Quantification of Nrf2 level in nucleus and cytosol fraction, respectively. Intensity of bands on Western blots was analyzed using a ImageJ software. Data are presented as Means ± S.E.M of three independent experiments. * *p* < 0.05 vs. hemin and ** *p* < 0.005 vs. control. ^#^
*p* < 0.01 vs. control and ^##^
*p* < 0.005 vs. hemin. Arrow indicates Nrf2 molecule specific bands.

**Figure 8 medicina-56-00268-f008:**
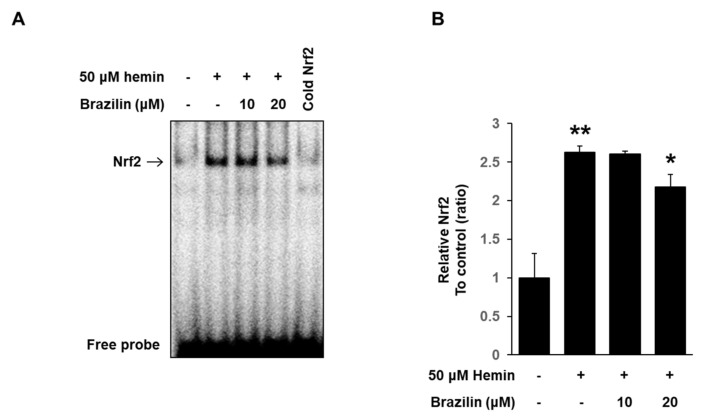
Effect of brazilin on Nrf2 DNA binding activity to murine HO-1 promoter oligonucleotide. (**A**) Effects of brazilin on Nrf2 DNA binding activity on MCF-7 cells. Brazilin were treated with cells in the presence of hemin. Levels of Nrf2 DNA binding was analyzed by EMSA following Materials and Methods. (**B**) Quantification of Nrf2 complex on EMSA data was measured using ImageJ software. Data are presented as Means ± S.E.M of three independent experiments. * *p* < 0.05 vs. hemin and ** *p* < 0.001 vs. control.

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
