# Peer review of "Roles of JNK/Nrf2 Pathway on Hemin-Induced Heme Oxygenase-1 Activation in MCF-7 Human Breast Cancer Cells"

_medicina, 2020, doi:10.3390/medicina56060268_

Round 1
Reviewer 1 Report
The manuscript by Jang et al. described that brazilin, a nature product from Caesalpinia sappan, upregulates HO-1 through JNK activation.
There are some concerns needed to be addressed by the authors.
- HO-1 has been identified that it exerts a protective role for cancer cells in response to chemotherapy. However, Hemin is not a chemotherapeutic agent. Brazilin-inhibited HO-1 is unable to correlate with anti-cancer effects.
- That Brazilin induces HO-1 has been described in other cell types. Whether Brazilin alone can increase HO-1 in MCF-7 cells?
- ERK inhibitor, PD98059, has no effect on Brazilin-induced HO-1. It is unable to conclude that Brazilin regulates hemin-induced HO-1 activation through phosphorylation of Erk.
- Please list the catalog number of antibodies in the manuscript.
- The symbol indicating the statistical significance is not appropriated. Please correct them.
Author Response
May 12, 2020
Manuscript number: Medicina-787954
Title: Roles of JNK/Nrf2 pathway on hemin-induced heme oxygenase-1 activation in MCF-7 human breast cancer cells
Authors: Hye-Yeon Jang, Kwang-Hyun Park*, and Jong-Suk Kim*
We appreciated reviewer’s pertinent comments. We revised our manuscript responding to the reviewer’s comments. The reviewers pointed out that more description in the paper were required for supporting the hypothesis. Therefore, we arranged data in figure and described answers to reviewer’s comments in enclosures.
We would be happy if you would reconsider our manuscript to publish in the ‘Medicina’.
Sincerely yours.
Jong-Suk Kim, M.D., Ph.D
Department of Biochemistry,
Chonbuk National University Medical School, Jeonju, 54914, Republic of Korea.
Tel: 82-63-270-3085, Fax: 82-63-274-9833.
E-mail: jsukim@chonbuk.ac.kr
Kwang-Hyun Park. Ph.D
Department of Emergency Medical Rescue and Department of Oriental Pharmaceutical Development, Nambu University, Gwangju, 62271, Republic of Korea
Tel/Fax. +82-62-970-0220
E-mail: khpark@chonbuk.ac.kr
Enclosures
Response to Reviewer’s Comments
Response to Reviewer’s Comments
<Reviews 1>
The manuscript by Jang et al. described that brazilin, a nature product from Caesalpinia sappan, upregulates HO-1 through JNK activation.
There are some concerns needed to be addressed by the authors.
- HO-1 has been identified that it exerts a protective role for cancer cells in response to chemotherapy. However, Hemin is not a chemotherapeutic agent. Brazilin-inhibited HO-1 is unable to correlate with anti-cancer effects.
<Responses> Yes. As your comments, hemin is not a chemotherapeutic agent. On the contrary, hemin is available HO-1 inducer in various cancer cell lines. Therefore, it would be a nice choice to examine that regulation of HO-1-mediated cancer progression with various candidates of chemotherapeutic agent. To support this point, more descriptions have been added to the discussion section.
- That Brazilin induces HO-1 has been described in other cell types. Whether Brazilin alone can increase HO-1 in MCF-7 cells?
<Responses> No. As reviewer’s comments, we also investigated whether brazilin-induced HO-1 expression, but it was not observed (data not shown).
- ERK inhibitor, PD98059, has no effect on Brazilin-induced HO-1. It is unable to conclude that Brazilin regulates hemin-induced HO-1 activation through phosphorylation of Erk.
<Responses> Yes. As reviewer’s comments, we considered statistic analysis and corrected mistakes in Fig 6C. Thank you for the opportunity to correct the data accurately by your exact point.
- Please list the catalog number of antibodies in the manuscript.
<Responses> Yes. As reviewers comments, we have been list the catalog number of antibodies in the manuscript.
- The symbol indicating the statistical significance is not appropriated. Please correct them.
<Responses> Yes. As reviewer’s comments, we corrected to appropriated symbols indicating the statistical significance following 'Instructions for author'.

Reviewer 2 Report
The manuscript presented by Jang and co-authors describes a study about the effect of hemin in heme oxygenase-1 expression in human breast cancer cells, MCF-7. Although the effects of hemin in the HO-1 expression is known for other cell lines, this study also includes the observation of JNK signaling pathway and Nrf2 transcription factor. Authors also suggest that HO-1 is an important anticancer-target of brazilin in human breast cancer as it inactivates JNK/Nrf2 pathways in MCF-7 cells.
In general terms, the results of the experiments are well presented and some of the conclusions are quite straight forward.
Nevertheless, there are a few main questions that should be addressed:
- Although the number of MCF-7 cells used in each assay are presented in the methods section, no comment is done about the cell growth phase, especially what after the 24h incubation periods. Are the cells in the lag or exponential growth phase? Is this relevant for the assays? How should this relate with the incubation times used? Is the starting number of cells, sometimes more than an order of magnitude difference, relevant for the assays performed?
- In figure 7, authors demonstrate the rffect of brazilin on hemin-induced Nrf2 activation, namely how it regulates its translocation into nuclear region. However, by observation of panel A and B, no significant difference is observed between the samples treated with hemin and hemin+brazilin. On the contrary, in panel C, there is a correlation with the brazilin amount. Therefore, authors should comment on this result in a clear way. Besides, panel A should be reformulated in order to show clearly the Nrf2 cytosol section as a second band seems to be present but it is cut in the current version.
- The discussion section should be revised in order to present a proper integration of the work presented. The results are presented throughout the manuscript without a clear description of the observations and then, they are merely stated in the discussion section. As such, it is hard to evaluate and visualize the conclusions proposed.
Besides these points, authors should revise Figure’s 5 caption as it states that “Brazilin regulates hemin-induced HO-1 activation through JNK/Nrf2 signaling pathway 179 in MCF-7 cells”, although no brazilin is mentioned in the figure itself. Was it used? If so, the assay conditions should be stated.
In addition, I believe that manuscript needs to be revised by an English-speaking native.
Author Response
May 12, 2020
Manuscript number: Medicina-787954
Title: Roles of JNK/Nrf2 pathway on hemin-induced heme oxygenase-1 activation in MCF-7 human breast cancer cells
Authors: Hye-Yeon Jang, Kwang-Hyun Park*, and Jong-Suk Kim*
We appreciated reviewer’s pertinent comments. We revised our manuscript responding to the reviewer’s comments. The reviewers pointed out that more description in the paper were required for supporting the hypothesis. Therefore, we arranged data in figure and described answers to reviewer’s comments in enclosures.
We would be happy if you would reconsider our manuscript to publish in the ‘Medicina’.
Sincerely yours.
Jong-Suk Kim, M.D., Ph.D
Department of Biochemistry,
Chonbuk National University Medical School, Jeonju, 54914, Republic of Korea.
Tel: 82-63-270-3085, Fax: 82-63-274-9833.
E-mail: jsukim@chonbuk.ac.kr
Kwang-Hyun Park. Ph.D
Department of Emergency Medical Rescue and Department of Oriental Pharmaceutical Development, Nambu University, Gwangju, 62271, Republic of Korea
Tel/Fax. +82-62-970-0220
E-mail: khpark@chonbuk.ac.kr
Enclosures
Response to Reviewer’s Comments
Response to Reviewer’s Comments
<Reviews 2>
The manuscript presented by Jang and co-authors describes a study about the effect of hemin in heme oxygenase-1 expression in human breast cancer cells, MCF-7. Although the effects of hemin in the HO-1 expression is known for other cell lines, this study also includes the observation of JNK signaling pathway and Nrf2 transcription factor. Authors also suggest that HO-1 is an important anticancer-target of brazilin in human breast cancer as it inactivates JNK/Nrf2 pathways in MCF-7 cells.
In general terms, the results of the experiments are well presented and some of the conclusions are quite straight forward.
Nevertheless, there are a few main questions that should be addressed:
- Although the number of MCF-7 cells used in each assay are presented in the methods section, no comment is done about the cell growth phase, especially what after the 24h incubation periods. Are the cells in the lag or exponential growth phase? Is this relevant for the assays? How should this relate with the incubation times used? Is the starting number of cells, sometimes more than an order of magnitude difference, relevant for the assays performed?
- <Responses> Yes. Thank you very much reviewer’s exact points. As reviews comments, cell culture and viability test paragraph were corrected in Materials and Methods.
- In figure 7, authors demonstrate the rffect of brazilin on hemin-induced Nrf2 activation, namely how it regulates its translocation into nuclear region. However, by observation of panel A and B, no significant difference is observed between the samples treated with hemin and hemin+brazilin. On the contrary, in panel C, there is a correlation with the brazilin amount. Therefore, authors should comment on this result in a clear way. Besides, panel A should be reformulated in order to show clearly the Nrf2 cytosol section as a second band seems to be present but it is cut in the current version.
- <Responses> As reviewer’s comment’s, less significant difference is observed between hemin and hemin+brazilin group. Indeed, a larger molecule second band was observed in the nuclear fraction but smaller molecule bands observed in the cytosol fraction. Therefore, arrows were indicated the Nrf2 molecule specific bands in Fig 7A.
- The discussion section should be revised in order to present a proper integration of the work presented. The results are presented throughout the manuscript without a clear description of the observations and then, they are merely stated in the discussion section. As such, it is hard to evaluate and visualize the conclusions proposed.
- <Responses> Thanks for the comments. As you pointed out, we corrected discussion section in the revised manuscript
- Besides these points, authors should revise Figure’s 5 caption as it states that “Brazilin regulates hemin-induced HO-1 activation through JNK/Nrf2 signaling pathway 179 in MCF-7 cells”, although no brazilin is mentioned in the figure itself. Was it used? If so, the assay conditions should be stated. In addition, I believe that manuscript needs to be revised by an English-speaking native.
- <Responses> Yes. We revise again by English native speaker.
- <Responses> Yes. As reviews questions, Fig 5 presented that effect of hemin on JNK/ERK/p38 activities with specific inhibitors but not used brazilin. Therefore, the title of Fig 5 was corrected in revised manuscripts.

Reviewer 3 Report
The manuscript by Jang et al. describes the in vitro effect exerted by brazilin to block hemin-induced HO-1 expression in MCF-7 cells through inactivation of JNK/Nrf2 in MCF-7 cell. Overall the manuscript is well written, and the description of the experimental part is clear. Some criticism is about the lack of reference compounds used as a control during the in vitro assays. For instance, the use of a HO-1 inhibitor, such as ZnPP or SnPP, would be have been ideal.
My suggestions are as follows:
- The style of the two chemical structures is not homogeneous; thus the use of appropriate software to draw them is recommended.
- The introduction may be implemented. For example, the authors should better describe the significance of the use of HO-1 inhibitor/inducer to treat cancer and other diseases. The authors should report more recent works to provide an up to date information on the field. For your reference, a shortlist of recent studies is listed bellow:
- Ciaffaglione et al. Int J Mol Sci 2020 Mar 11;21(6). pii: E1923. doi: 0.3390/ijms21061923
- Sorrenti et al. Int J Mol Sci. 2019 May 17;20(10). pii: E2441. doi: 10.3390/ijms20102441
- Podkalicka et al. Biomolecules. 2020 Jan; 10(1): 143. Published online 2020 Jan 16. doi: 10.3390/biom10010143
- Lines 232: the authors have mentioned the HO-2, however, the description of this isoform has been wholly neglected. A brief description of this isoform should be added along with appropriate references (i.e., suggested review articles: doi: 10.1016/j.ejmech.2019.111703; doi: 10.1155/2014/604981).
- Discussion and conclusion may be collapsed into one section and could be implemented.
Author Response
May 12, 2020
Manuscript number: Medicina-787954
Title: Roles of JNK/Nrf2 pathway on hemin-induced heme oxygenase-1 activation in MCF-7 human breast cancer cells
Authors: Hye-Yeon Jang, Kwang-Hyun Park*, and Jong-Suk Kim*
We appreciated reviewer’s pertinent comments. We revised our manuscript responding to the reviewer’s comments. The reviewers pointed out that more description in the paper were required for supporting the hypothesis. Therefore, we arranged data in figure and described answers to reviewer’s comments in enclosures.
We would be happy if you would reconsider our manuscript to publish in the ‘Medicina’.
Sincerely yours.
Jong-Suk Kim, M.D., Ph.D
Department of Biochemistry,
Chonbuk National University Medical School, Jeonju, 54914, Republic of Korea.
Tel: 82-63-270-3085, Fax: 82-63-274-9833.
E-mail: jsukim@chonbuk.ac.kr
Kwang-Hyun Park. Ph.D
Department of Emergency Medical Rescue and Department of Oriental Pharmaceutical Development, Nambu University, Gwangju, 62271, Republic of Korea
Tel/Fax. +82-62-970-0220
E-mail: khpark@chonbuk.ac.kr
Enclosures
Response to Reviewer’s Comments
Response to Reviewer’s Comments
<Reviews 3>
The manuscript by Jang et al. describes the in vitro effect exerted by brazilin to block hemin-induced HO-1 expression in MCF-7 cells through inactivation of JNK/Nrf2 in MCF-7 cell. Overall the manuscript is well written, and the description of the experimental part is clear. Some criticism is about the lack of reference compounds used as a control during the in vitro assays. For instance, the use of a HO-1 inhibitor, such as ZnPP or SnPP, would be have been ideal.
My suggestions are as follows:
- The style of the two chemical structures is not homogeneous; thus the use of appropriate software to draw them is recommended.
- <Responses> Yes. As reviewer’s comments, we cited high quality images.
- The introduction may be implemented. For example, the authors should better describe the significance of the use of HO-1 inhibitor/inducer to treat cancer and other diseases. The authors should report more recent works to provide an up to date information on the field. For your reference, a shortlist of recent studies is listed bellow:
- Ciaffaglione et al. Int J Mol Sci 2020 Mar 11;21(6). pii: E1923. doi: 0.3390/ijms21061923
- Sorrenti et al. Int J Mol Sci. 2019 May 17;20(10). pii: E2441. doi: 10.3390/ijms20102441
- Podkalicka et al. Biomolecules. 2020 Jan; 10(1): 143. Published online 2020 Jan 16. doi: 10.3390/biom10010143
<Responses> Thanks for the comments. As you pointed out, we cited more recent works in introduction section of revised manuscript
- Lines 232: the authors have mentioned the HO-2, however, the description of this isoform has been wholly neglected. A brief description of this isoform should be added along with appropriate references (i.e., suggested review articles: doi: 10.1016/j.ejmech.2019.111703; doi: 10.1155/2014/604981).
- <Responses> As you pointed out, we cited current reference in introduction section in the revised manuscript
- Discussion and conclusion may be collapsed into one section and could be implemented.
- <Responses> Thanks for the comments. Discussions and conclusion sections have been reduced to one section.

Round 2
Reviewer 1 Report
- This study demonstrated that Brazilin downregulates Hemin-induced HO-1 via the JNK-Nrf2 pathway. Brazilin has a significant decrease in Hemin-induced HO-1. However, both Brazilin and Hemin had no effect on the growth of MCF-7 breast cancer cells. Thereby, it is unable to suggest that Brazilin exerts the anti-cancer activity and HO-1 contributes to the growth of MCF-7 breast cancer cells. The description of Line 51-53 should be modified.
- The presentation of statistics is not appropriated in Figure 4, Figure 5, and Figure 6. Please correct them.
Author Response
Response to Reviewer’s Comments
<Reviews 1>
- This study demonstrated that Brazilin downregulates Hemin-induced HO-1 via the JNK-Nrf2 pathway. Brazilin has a significant decrease in Hemin-induced HO-1. However, both Brazilin and Hemin had no effect on the growth of MCF-7 breast cancer cells. Thereby, it is unable to suggest that Brazilin exerts the anti-cancer activity and HO-1 contributes to the growth of MCF-7 breast cancer cells. The description of Line 51-53 should be modified.
<Responses> Yes. As reviewer’s comments, we modified unclear descriptions of 2nd paragraph in Introduction. The revised part is indicated in blue font.
- The presentation of statistics is not appropriated in Figure 4, Figure 5, and Figure 6. Please correct them.
<Responses> Yes. All presentation of statistics were corrected to appropriated description in Figure legends.

Reviewer 2 Report
The quality of the revised version of the manuscript presented by Jang and co-authors improved with the modifications performed.
The authors addressed most of the recommendations, namely in terms of data presentation and methodology, which translated in clearer presentation of the results.
Nevertheless, the integration of the results remains superficial in discussion section, which impairs a proper understanding of the outcomes of the work. A proper revision of the text should be performed, namely in the new parts added.
Author Response
Response to Reviewer’s Comments
<Reviews 2>
The quality of the revised version of the manuscript presented by Jang and co-authors improved with the modifications performed.
The authors addressed most of the recommendations, namely in terms of data presentation and methodology, which translated in clearer presentation of the results.
Nevertheless, the integration of the results remains superficial in discussion section, which impairs a proper understanding of the outcomes of the work. A proper revision of the text should be performed, namely in the new parts added.
<Responses> Yes. As reviewer’s comments, we edited discussion section for proper understanding of the outcomes of our data. The revised part is indicated in blue font.

Round 3
Reviewer 1 Report
I have no further questions for the authors
This manuscript is a resubmission of an earlier submission. The following is a list of the peer review reports and author responses from that submission.